# Towards Development of Specular Reflection Vascular Imaging

**DOI:** 10.3390/s22082830

**Published:** 2022-04-07

**Authors:** Timothy Burton, Gennadi Saiko, Alexandre Douplik

**Affiliations:** 1Department of Biomedical Engineering, Ryerson University, Toronto, ON M5B 2K3, Canada; timothy.burton@ryerson.ca; 2Department of Physics, Ryerson University, Toronto, ON M5B 2K3, Canada; gsaiko@ryerson.ca; 3iBest, Keenan Research Centre of the LKS Knowledge Institute, St. Michael Hospital, Toronto, ON M5B 1T8, Canada

**Keywords:** biomedical imaging, jugular vein, carotid artery, pulse propagation

## Abstract

Specular reflection from tissue is typically considered as undesirable, and managed through device design. However, we believe that specular reflection is an untapped light-tissue interaction, which can be used for imaging subcutaneous blood flow. To illustrate the concept of subcutaneous blood flow visualization using specular reflection from the skin, we have developed a ray tracing for the neck and identified conditions under which useful data can be collected. Based on our model, we have developed a prototype Specular Reflection Vascular Imaging (SRVI) device and demonstrated its feasibility by imaging major neck vessels in a case study. The system consists of a video camera that captures a video from a target area illuminated by a rectangular LED source. We extracted the SRVI signal from 5 × 5 pixels areas (local SRVI signal). The correlations of local SRVIs to the SRVI extracted from all pixels in the target area do not appear to be randomly distributed, but rather form cohesive sub-regions with distinct boundaries. The obtained waveforms were compared with the ECG signal. Based on the time delays with respect to the ECG signal, as well as the waveforms themselves, the sub-regions can be attributed to the jugular vein and carotid artery. The proposed method, SRVI, has the potential to contribute to extraction of the diagnostic information that the jugular venous pulse can provide.

## 1. Introduction

Bioimaging is an important domain of clinical diagnostics. Optical bioimaging is a superior modality for surface surveillance of various physiological parameters, such as those related to blood supply and flow. Monitoring of these hemodynamic parameters is critical for cardiovascular disease diagnostics, with this family of diseases being a main cause of death in Western countries such as the United States [1].

### 1.1. Light-Tissue Interactions

The interaction of light with tissue enables a suite of invaluable medical diagnostic tools. Light shone on tissue results in a variety of possible interactions, all of which depend on both the properties of the tissue and the light. Firstly, light can be absorbed, which is the transference of energy from the light to the tissue. Secondly, light can be scattered elastically within the tissue, which is the re-emission of light from the scattering particles without loss of energy, but in a different direction. This interaction is referred to as diffuse reflectance or diffuse backscattering. Thirdly, light can be scattered inelastically, which is known as Raman scattering, in which the scattered light has a different energy from the incident light. Fourthly, light can be absorbed and re-emitted with a different energy, which is referred to as fluorescence, or more broadly, as luminescence. Finally, light can be specularly reflected directly from the surface of the tissue in a mirror-like reflection. Specular reflection, also known as Fresnel reflection, is caused by a mismatch in refraction index on the interface between the tissue and the air. For clarity, terminology in the literature can vary; oftentimes, the Fresnel reflectance from smooth surface is called specular reflection, and from rough surface, diffuse reflectance. However, to avoid confusion with diffuse backscattering within the tissue (mentioned here as the second light-tissue interaction), our meaning of specular reflection is the Fresnel reflection from any type of surface. Specifically, specular reflection may come from a smooth or rough surface. The “diffuse reflection” term is therefore reserved for the second light-tissue interaction mentioned here, which is diffuse back-scattering from within tissue.

All these interactions may occur simultaneously and depend on the tissue’s optical properties. The first four light-tissue interactions are employed in a variety of existing medical diagnostic applications [2,3,4,5]. In contrast, the last, specular reflection, is typically considered as undesirable, and its effect reduced to tolerable levels through device design. 

### 1.2. Specular Reflection in Photoplethysmography

In this paper, we will focus on optical methods that can provide essential information about hemodynamic parameters and heart function. For instance, photoplethysmography (PPG) devices aim to capture a waveform primarily indicative of changes in blood volume in tissue and can be used to detect heart rate or calculate blood oxygen saturation [2]. The PPG signal embeds influence from both the arterial system (oscillatory in nature) and the venous system (slow-moving in nature). The arterial oscillatory component is driven by ejection of blood from the left ventricle of the heart to the systemic circulation in the body. The rise of the PPG waveform is associated with cardiac systole, peaking at maximal volume, and the fall of the waveform with cardiac diastole [6]. The PPG also exhibits a diastolic peak during the falling component of the waveform associated with pulse wave reflections [6]. Clinical PPG devices are typically transmission mode, in which light passes through the tissue, undergoing absorption and scattering, before being detected on the other side of the tissue. Typically, transmission mode PPG devices are in the form of a clip, such that mechanical force created by a spring fixes the light emitter and detector in a stable position on opposite sides of the tissue [7]. Transmission mode is therefore limited to peripheral tissue where such a clip can be attached, such as the finger, toe, or ear. It is important to maintain robust skin contact when capturing PPG signals using transmission mode devices. Transient changes in skin contact may alter the signal; for example, emitter movement may cause the light to strike the tissue at an angle that causes increased specular reflection from the skin, and therefore, less light enters the tissue.

Reflection mode PPG devices [8] also exist for more specialized clinical applications, such as brachial-ankle pulse wave velocity [3] and neo-natal monitoring [9], as well as consumer devices such as fitness trackers and smartwatches. In this case, both the light emitter and detector are positioned against the same side of the tissue. This configuration eliminates the dependency of PPG on the geometry of peripheral tissue, enabling measurement of practically any body surface location [8]. The light emitted from the device can be reflected from the surface of the skin (i.e., specular reflection), or it penetrates the tissue and interacts with it through multiple absorption and scattering events prior to capture by the detector due to the backscattering (i.e., diffuse reflectance). However, reflection mode PPG devices are more susceptible to specular reflection than transmission mode PPG devices due to relative proximity of the emitter and detector. Reflection PPG devices must therefore be thoughtfully designed as to minimize the impact of specular reflections, which, in this case, are a source of noise that may contaminate the signal. Design options to overcome this noise source include optically shielding the light source from the detector, such that skin specular reflections that have high angular density do not enter the detector and overwhelm the relatively lower-strength scatter signal [10]. Remote PPG (rPPG) is an extension of reflection mode PPG, such that the device is not required to be in contact with the skin. rPPG is increasing in popularity since the non-contact property enables applications in telehealth and sensitive skin assessment, as well as the ability to simultaneously capture multiple points of measurement that can be displayed as an image. However, rPPG is more vulnerable to specular reflection than the base reflection mode PPG due to increased difficulty of integrating optical shielding. 

As described, specular reflection is problematic for PPG, as it is for most optical diagnostic applications. Interestingly, specular reflection is disruptive while being relatively weak in comparison to diffuse reflectance (such as that measured in reflection mode PPG and rPPG), which is typically around 20–60% (depending on the wavelength), in comparison to only several percent for specular reflection [11]. This strength relationship holds regardless of whether the light source is collimated or diffuse. The outsized effect of specular reflection as compared to diffuse reflection is explained by the concentration of specular reflection within a narrow solid angle, whereas diffuse reflection is distributed across a large (2π) angle. Specifically, at angles with high specular reflectance and average diffuse reflection, specular reflection can overwhelm diffuse reflection. When occurring in imaging applications (such as rPPG), this phenomenon is known as “hot-spots” [12] on images, where specular reflection locally saturates the image sensor. Despite the issues specular reflection may cause, it is not insurmountable; in addition to the control measures already described, polarization gating is a practical approach that can be used to both identify and isolate specular reflection [13,14]. 

### 1.3. Motivation for Present Work

As described, cardiovascular disease is a main cause of death in the developed world. While technology has advanced substantially to aid clinicians in the diagnosis and management of cardiovascular disease, there is little availability of assistive devices to extract the jugular venous pulse (JVP) waveform. The JVP waveform is the signal produced by venous return from systemic circulation in the head and neck to the right atrium. The JVP waveform embeds a wealth of information, and contains two primary peaks. The first peak is caused by atrial contraction followed by right ventricular contraction, and the second peak is caused by right atrial pressure release through the opening of the tricuspid value [15]. The JVP signal is a valuable waveform, informing a spectrum of disease states, such as tricuspid stenosis and regurgitation [16], and atrial fibrillation [16]. Unfortunately, the assessment of the jugular venous signal hand during a routine clinical exam is difficult, necessitating the differentiation between jugular and carotid pulsations [17]. Due to this difficulty to perform, it is challenging for new physicians to achieve competence in this skillset; thus, assistive devices to extract this waveform would be of clinical utility. Some initial work has demonstrated the viability of optical approaches; specifically, reflection mode PPG has been used to extract the JVP waveform [15]. However, reflection mode PPG has key limitations associated with signal strength, confounders, and susceptibility to operator error. Contact PPG methods, including reflection mode PPG, are inherently restricted to a single point of measurement and therefore rely on the operator to place the device in a precise location to measure the desired signal. For instance, in [15], the investigators had to use an infrared vein camera to achieve appropriate placement of the PPG device on the neck. Extension from reflection mode PPG to rPPG addresses the single point of measurement limitation, but rPPG exhibits the remaining limitations associated with PPG. 

As mentioned, PPG is limited by signal strength, which can be understood through the origin of the signal. PPG signals originate from 1.5–2 mm below the surface of the skin [18], and photons must transit that distance, including multiple attenuation events such as scattering, to emerge from the skin to be captured by the detector. Therefore, signal strength is an inherent limitation of PPG caused by the source of the signal itself. PPG properties can also be affected by confounders that differ across patients. Specifically, rPPG signal strength is reduced in the presence of high melanin concentration, and therefore rPPG performs more poorly on darker skin tones [19]. This limitation has the potential to contribute to racial/ethnic health care disparities [20]. A similar effect has been observed by the presence of cosmetic makeup [21], which can reasonably be expected to be worn by a large proportion of patients. 

Therefore, we believe that while the reflection mode PPG methodology described in the literature for extraction of JVP waveforms is an important step, and the limitations associated with PPG call for a new modality to meet this need. 

### 1.4. Specular Reflection Vascular Imaging

Therefore, we propose a new imaging modality, specular reflection vascular imaging (SRVI). We believe that in contrast to specular reflection’s typical undesirable role, it is an untapped light-tissue interaction, and in particular, that it has value for imaging subcutaneous blood flow. Specifically, the pulse wave is known to cause elastic deformation [22]. It has been proposed that the deformation extends to the connective tissue components of the dermis, resulting in changing optical properties that contribute to the origin of the PPG signal [22]. We further hypothesize that the pressure caused by the mechanical pulse wave propagates to the very surface of the skin and that the skin surface deforms in a manner reflective of the underlying blood flow. We believe that the skin surface deformation may be detectible through changes in specular reflection that can be captured with camera. Further, capturing the signal in this manner addresses the limitations of PPG that we have described. To explore this hypothesis, we will present a theoretical justification for SRVI. In addition, to demonstrate practical feasibility of this methodology, we performed a case study which acquired initial waveforms from the carotid and jugular vessels towards a “proof of principle”. 

## 2. Theory

### 2.1. Specular Reflection

As previously described, specular reflection is one of the primary types of light-tissue interaction. Specular reflection is weakly dependent on the wavelength of incident light and can therefore occur in a similar manner across the spectrum. For the normal incidence, the coefficient of specular reflection *R_s_* will depend on the relative index of refraction *n*:(1)$$R_{s}=\frac{{\left(n-1\right)}^{2}}{{\left(n+1\right)}^{2}}$$

Typical values of *R_s_* are relatively low, in the range of a few percent, because typical tissue will exhibit *n* in the range of 1.33–1.45 [11]. Despite this low value of specular reflection, such measurement is problematic as a noise source for the other light-tissue interactions due to its propensity to create hot-spots, as previously discussed. 

Specular reflection is dependent not only on refractive index but also on the roughness of the surface. Typical surfaces are not perfectly smooth and therefore exhibit some roughness through surface components meeting at varying angles (per Figure 1a), where the size of the components dictates the roughness of the surface from the smooth surface to closer to Lambertian surface [23]. The variation in angles causes angular distribution in specular reflection, as is common in matte surface finishing of metal or plastic. Surfaces that may visually appear to be smooth may contain roughness that is not visually perceptible, such as skin. For specular reflection to be maximal in a given angle of reflection, the surface must be as smooth as possible, per Figure 1b.

The smooth surface component of the specular reflection can be increased in through application of a “smoothening” agent such as oil, which typically functions to populate the skin concavities (such as in Figure 1a) and generate a smooth surface (such as in Figure 1b). 

### 2.2. Model

The following model describes how subcutaneous blood flow may be observable using a non-invasive sensing methodology based on the hypothesized specular reflection mechanism, using the neck as the initial anatomical location. 

In this model, the neck is a cylinder with approximate radius r = 5–6 cm [24] (with women having lower radius and men higher), with a corresponding circumference of 31–38 cm. The height of the cylinder is h = 10–11 cm [25], again representing the variability between genders (lower for women, higher for men). Further, the neck is illuminated by a rectangular light source with a height l and width 2*w*, which is aligned with the direction of the neck (i.e., cylinder axis is parallel to height dimension of the light source). Next, L and w are derived. The light source is located on the distance P from the neck. Then, the neck can be considered as a cylindrical convex mirror. The ray tracing for such mirror is depicted in Figure 2.

From the elementary optics [26], we can draw several observations:(1)The rays are always divergent, so the image is virtual and located on the opposite side of the mirror from the source.(2)If the source is located at distance *P* from the mirror, then the image is located at distance *Q*, which is given by the mirror formula (here all distances are positive)
(2)$$\frac{1}{P}-\frac{1}{Q}=-\frac{2}{r}$$

(3)The magnification of the image, *m,* is given by the following expression:


(3)
$$m=\frac{Q}{P}$$


Thus, the image of the light source with height l and width 2*w* is a rectangle with height l and width 2*w*′, which is located on the distance *Q* inside the mirror. Here, *Q* can be found from Equation (2). The semi-width *w*′ can be found from Equation (4) as:(4)$$w^{\prime }=wm=w\frac{1}{1+2P/r}$$

For example, for realistic *r* = 6 cm and *P* = 12 cm, then *Q* = 2.4 cm and *m* = 0.2. Thus, the observer will see a bright rectangle the height l and width 2*w*′, which is located at distance *Q* inside the mirror. Deviation of the image will reflect changes in the geometry of the reflecting mirror, i.e., the skin of the neck.

The geometry of the experiment is depicted in Figure 3. The observer (i.e., camera) is located on the distance *L* from the neck. To image the target area with height *H* and width *W*, the height of the light source *l* can be found from the following inequality: (5)$$\frac{H}{L}\leq \frac{l}{L+Q}=\frac{l}{L+\frac{Pr}{r+2P}}$$

Similarly, for the width: (6)$$\frac{W/2}{L}\leq \frac{w^{\prime }}{L+Q}=\frac{wr}{L\left(r+2P\right)+Pr}$$

If the light source, height and width are smaller than the value provided by the equivalence in Equations (5) and (6), then its image is less than the required target area. If it is larger, then some parts of the light source image will be cropped; however, it will cover the full target area. Further, the observer’s field of view (FOV) should accommodate the required target area.

Thus, in the case of an ideal cylinder, the observer will see a bright rectangle with height l and width 2*w*′ located at the distance *L* + *Q*. Each point of this rectangle will correspond to one point on the surface of the cylinder (1-to-1 mapping). Deviations from the ideal cylindrical geometry (caused by blood volume changes) will result in changes to the brightness at a particular point of the rectangle.

In reality, the neck is not perfectly cylindrical, which will result in inhomogeneous light intensity within the bright rectangle. Note that the source and images are both rectangles. Per our stated hypothesis, the inhomogeneous light intensity captures spatially varying specular reflection caused by the skin deformation, which is in turn caused by the pressure from the mechanical pulse wave propagating to the surface of the skin. 

Variations in the location of the vessels of interest are common [27] but lie approximately as shown in Figure 4 (where the jugular and carotid are between 0.75–1.16 cm and 1.57–1.82 cm, respectively, from the skin, depending on inferior/superior position and bilateral variation [28]); therefore, the imaging region of interest (ROI) was set to have a conservative width (to accommodate anatomical variations) of *W* = 2.5–3 cm. Thus, using Equation (6), we can calculate the required semi-width of the light source:(7)$$w\geq \frac{W}{2}\left(1+\frac{2P}{r}+\frac{P}{L}\right)$$

For example, for *W* = 3 cm (3 cm wide target area), *r* = 5 cm (using the female neck radius as the worst case) and *L* = *P* =12 cm, then *w* must be at least 10.2 cm. Thus, for proper imaging of the 3 cm wide area, we need a light source with height *l* = 12 cm and width 2*w* = 20 cm.

### 2.3. Translation from Theoretical to Experimental 

Let us consider a vessel, which transports blood subcutaneously. For simplicity, assume that during cardiac diastole, the skin above the vessel is flat. During systole, the increase in the diameter of the vessel due to pulse wave propagation is translated into a radial displacement of the tissue surrounding the vessel. The displacement propagates to the skin, given sufficient proximity to the skin and pressure in the vessel. Assume that the maximum skin displacement, occurring in the nearest skin area to the vessel (i.e., directly over vessel), is Δz. The displacements over nearby skin areas will be in the [0, Δz] range. However, the displacement in the nearest skin area results into changes to the slope of the skin with respect to the initial flat orientation. The change in slope spatially ranges from zero for remote, unaffected skin areas, to a maximum value, and then it decreases to zero again in the nearest skin area over the vessel. 

Therefore, a particular region of skin, neither remote nor the nearest skin area, receives a non-zero slope due to the deformation caused by the vessel expansion. The result of the non-zero slope is reduced image brightness at that region of skin, as the corresponding specular reflection ray is reflecting at a different angle and migrating to a different area in the sensor plane. The distance of the ray displacement in the image for the fixed imaging geometry will be conditioned by the local slope. In other words, the effects of this on the image are a reduced registered intensity in the original spot and an increased value in another spot. If we consider reflection by all affected spots, then it will translate into temporal variations of light intensity in areas affected by mechanical displacements of the skin caused by pulse propagation. 

## 3. Materials and Methods

### 3.1. Data Collection

As a case study, we collected data from a healthy volunteer in a supine position. The study was approved by the Ryerson REB. A smoothening agent was applied to the region of interest (baby oil, Johnson & Johnson, New Brunswick, NJ, USA) to convert a rough surface into a smooth one, with the intention of maximizing the concentration of specular reflection within a narrow solid angle. The area of interest was illuminated by a rectangular LED broad-spectrum diffuse white light source (Neewer LED Video Light, Shenzen, China) with dimensions 23 × 20 cm. The light source was placed on the distance P ≃ 20 cm from the target area. Ambient room lighting was dimmed. The reflected image was captured using a scientific grade RGB CMOS camera (Basler acA2000-165uc; Ahrensburg, Germany) recording at 250 frames per second. The camera was located on the distance L ≃ 150 cm from the target area. A single-lead electrocardiogram (ECG) was acquired simultaneously with video data at a frequency of 400 Hz (Maxim MAX86150EVSYS; San Jose, CA, USA). Data were recorded for a total of 20 s, with 2 s removed due to the synchronization process, for a total of 18 s. The recorded time duration was selected to provide a reasonable number of cardiac cycles for analysis. 

### 3.2. Data Analysis

All analyses were performed using MATLAB R2020b (Mathworks; Natick, MA, USA). The data analysis outputs are described following and summarized in Table 1. 

First, the ROI was selected manually to be an area intended to capture the major neck vessels (carotid and jugular) and exclude any regions not of interest. For example, the chin is of no interest because it does not contain the target vessels, and non-tissue background pixels that occur the peripheral of the image are also of no interest. Therefore, any location within the ROI may possibly contain a signal from a target vessel. 

Next, filtering was performed, which is a standard initial operation when performing image processing. Gaussian filtering is a typical choice and was used herein to reduce noise in the video frames [29].

The global SRVI signal was then evaluated by averaging the intensities of all the pixels in the ROI at each time point, to create a three-channel (red, green, blue) time series at 250 Hz, resulting in 4500 total time points. While SRVI signals were evaluated for each color channel, the red channel was used for all subsequent analyses, as the signals from each channel capture similar information. Specifically, the refraction indices are close to each other; e.g., 1.405 and 1.386 at 460 and 630 nm, respectively [30], which results in 2.83% vs. 2.62% values of the coefficient of specular reflection. Local SRVI signals were then evaluated in each non-overlapping 5 × 5 pixels area within the ROI, such that all pixels in the ROI participated in one (and only one) local SRVI signal. The division into 5 × 5 regions was accomplished through indexing the matrix of pixel-level signals across the ROI—for example, the first local SRVI signal represents the first five rows and first five columns of pixels. 

Baseline wander is due to low-frequency artifacts not of interest to the analysis, and it may include a primary contribution from respiration in addition to other factors [31]. Baseline wander was removed by subtracting the 1 s moving average [32].

To visualize the spatial relationships of the local SRVI signals to the global signal, the Pearson correlation between each local SRVI signal and the global SRVI signal was calculated and displayed as a false color map overlaid on the first video frame. Similar methodology has been previously used for rPPG signals to determine whether a face presented to a facial recognition system is genuine or is a realistic mask attempting to trick the system [33]. In this case, cross-correlation between local rPPG signals was used as the metric, whereas Pearson correlation was used herein. 

Spatial clustering was performed to segment the ROI into two sub-regions that exhibited internally high-correlating SRVI signals, to identify regions of the neck that produce signals with high similarity. Similar methodology has previously been used in rPPG analysis, in that case to separate regions of tissue generating true rPPG signals from non-tissue regions generating only noise [34]. The SRVIs in each sub-region were averaged to generate representative SRVI signals. 

For visualization purposes, the SRVI and ECG signals were smoothed using 100 ms and 20 ms moving averages, respectively. 

## 4. Results

The ROI was selected manually to be an area intended to capture the major neck vessels (carotid and jugular), as shown in Figure 5. 

Figure 6a shows the spatial distribution of SRVI signals based on correlation to the global SRVI signal. The correlations of local SRVIs to the global SRVI do not appear to be randomly distributed but rather form cohesive sub-regions with distinct boundaries. The results of spatially clustering the ROI into cohesive sub-regions is shown in Figure 6b,c. Figure 6b represents the sub-region that is visually recognizable as highly correlating to the global SRVI signal, while Figure 6c represents the sub-region that ranges from low to negative correlation to the global SRVI signal. The sub-region in Figure 6b is positioned where the carotid artery is expected to be placed, with the top of the sub-region ending at the top of the ROI, close to the temporomandibular joint (also where the carotid is positioned, per Figure 4). The positioning of the internal and external jugular veins can vary with respect to the carotid artery but generally occurs just posterior to the carotid (see Figure 4), aligning with the sub-region identified in Figure 6c. 

Figure 7 plots the ECG (a) simultaneously captured with SRVI (b,c), with (d–f) demonstrating the synchronization of the signals on specific cardiac cycles from the complete acquisition. The peaks and waves mentioned in the following text are all marked in (d–f), with the following text referring to the symbol for each peak and wave at first mention. 

The red SRVI signal, representing the high-correlation sub-region, exhibits peaks in the region of the ECG t-wave (triangle). This is the expected property for an arterial signal [15], with the systolic peak (circle) close to the t-wave, indicating that this SRVI signal represents the carotid artery. The carotid SRVI signal shows rich dynamics, specifically the presence of a diastolic peak (star), which occurs approximately halfway between the offset of the t-wave and onset of the p-wave (star). The green curve represents the SRVI signal from the sub-region that ranges from low to negative correlation to the global SRVI signal. It has a primary peak that aligns with the ECG r-peak (circle) and a secondary peak occurring shortly after the t-wave. These waveform characteristics are indicative of the jugular venous pulse (JVP) [15], which is maximal in the region of the r-peak, indicating that the green SRVI signal originates from the jugular vein. Here, JVP peaks known as the *a* (star) and *c* (circle) peaks reflect atrial and right ventricular contraction, respectively; despite individual labelling, they are known to have very similar amplitude, often without a significant inflection occurring between them. The jugular SRVI signal agrees with this property, primarily demonstrating an elongated peak, as in (e,f), although more distinct *a* and *c* peaks are visible in (d). The third characteristic peak in the JVP waveform is the *v* peak (triangle), which occurs after the t-wave and captures maximal right atrial pressure prior to release by the tricuspid value [15]. The *v* peak is clearly visible in (d–f), occurring after each t-wave, preceding the next increase in amplitude towards the *a* peak occurring in the following cardiac cycle.

The alignment of the expected properties between the three signals (carotid SRVI, jugular SRVI and ECG) confirms that the SRVI signals truly represent the carotid and jugular vessels and, therefore, that the SRVI methodology appears to be capable of observing these vessels’ pulsation remotely. 

## 5. Discussion

We have presented the theoretical justification of a novel vascular imaging method, SRVI, and its initial implementation. It allowed us to collect spatial distribution of waveforms in a healthy volunteer in a case study.

The obtained waveforms were compared with ECG, which allowed us to attribute them to jugular vein and carotid artery. Thus, SRVI demonstrates the potential to extract JVP waveforms, which have significant clinical value. Cardiovascular disease is a significant burden in modern society, causing 23% of all deaths annually [1]; therefore, new tools for management of such disease are of value to assist clinicians in best diagnosing their patients. Utility of the JVP waveform was previously discussed, and it is of particular use in dilated cardiomyopathy, which is the thickening of the left ventricle preventing effective ejection of blood that can lead to heart failure [35]. The JVP signal has been used to identify two distinct subgroups of dilated cardiomyopathy: those with a dominant *a* wave with slow left and right ventricular filling times and those with a dominant *v* wave with preserved filling times [35]. Differing JVP properties across these dilated cardiomyopathy sub-types provide insight into the disease etiology and enable the clinicians to provide the most appropriate treatment. 

The proposed methodology has some interconnections with rPPG as well as with another method, laser speckle contrast imaging (LSCI). LSCI functions through quantification of the time variation of the random laser speckle pattern [36]. In this modality, statistical properties of the surface (or the scattering ensemble) can be derived from the light distribution in the far field. However, the difference arises from the structural randomness of the scatterer. While speckle formation deals usually with random structures, we have a perfectly arranged scatterer with some minor deviations. Thus, we consider the problem with the opposite degree of randomness.

The LSCI signal origin at ~0.5 mm [37] is not as deep as PPG (1.5–2 mm) but is still subcutaneous and can therefore be sensitive to melanin concentration [37]. While both rPPG and LSCI have limitations associated with increased concentration of melanin, and rPPG has also been shown to be impaired in the presence of cosmetic makeup, SRVI has the potential to overcome these issues. Specifically, specular reflection does not require skin penetration, and the specular reflection coefficient depends on the index of refraction only. This underlying mechanism of SRVI results in the strength and quality of the SRVI signal not being affected by these confounders. Further, PPG and LSCI originate from 0.5–2 mm below the surface of the skin, while in contrast, SRVI captures displacements of the surface of the skin as caused by pulse pressure wave, thus not requiring photons to propagate from subcutaneous depth. Therefore, SRVI has the potential to acquire a stronger signal than these modalities, even when they are operating under ideal conditions, due to the differing signal origins. 

Despite the described limitations of rPPG, the rPPG signal is present in the collected SRVI signal. Specifically, the current SRVI methodology does not explicitly exclude the contribution of the diffuse reflection signal. Thus, the SRVI signal is a weighted summation of the diffuse reflection signal from nearly arteries, and the specular reflection signal caused by skin displacement that we intend to capture. However, the diffuse reflection contribution is minimized through the SRVI methodology, which promotes the specular reflection through the use of the smoothening agent, targeted diffuse light source, and dimmed ambient lighting. Experimentally, the amplitude of SRVI was approximately 6-fold greater than a comparator rPPG dataset, demonstrating that the SRVI signal is dominated by specular reflection, with minimal contribution from diffuse reflection [38]. However, given the steps that must be taken to enhance specular reflection to the appropriate level, the SRVI acquisition process must be managed carefully, with the light source intensity and the exposure time of the camera selected accordingly to avoid saturation and overexposure. In particular, due to the changes to the scattering angle, the reflected light may move from one pixel to other, which may resemble constructive and deconstructive interference. Consequently, in the base case, the signal on the sensor must have sufficient margins and dynamic range (e.g., not to exceed 50% of saturation level).

The major limitation of the current work is the lack of understanding of SRVI subject variability since we have analyzed a single subject as a case study. Our immediate goal is to present a validation of the methodology on a large set of volunteers [38].

To further the development of SRVI, in future work, we plan to observe the waveforms of the jugular vein pulse in patients with elevated jugular vein pressure. Further, we also plan to perform a finite element analysis (FEA) simulation to better understand the sensitivity of SRVI to the relevant parameters, such as depth of the vessel from the surface of the skin, the blood flow rate, and pulse wave velocity. The use of collimated light also may reduce the role of diffuse reflection in the recorded signals. It will be explored in future work.

## Figures and Tables

**Figure 1 sensors-22-02830-f001:**
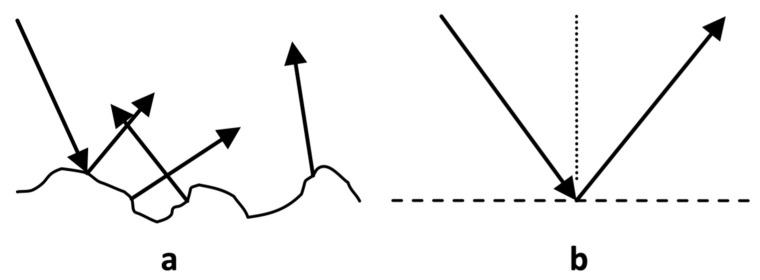
Two types of specular (Fresnel) reflection from tissue. Specular reflection from a rough surface (i.e., Lambertian reflectance) (**a**) and smooth surface (**b**).

**Figure 2 sensors-22-02830-f002:**
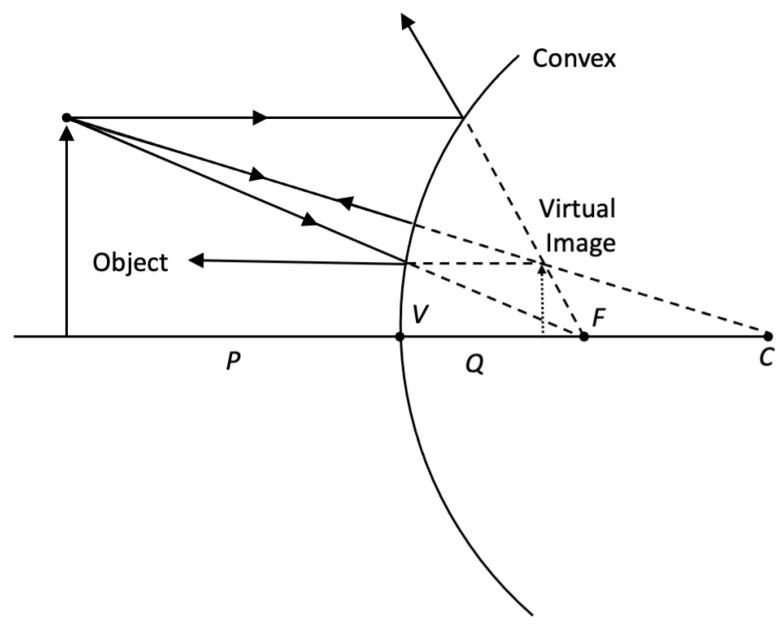
Graphical ray tracing for the convex cylindrical mirror (view from the axis of the cylinder).

**Figure 3 sensors-22-02830-f003:**
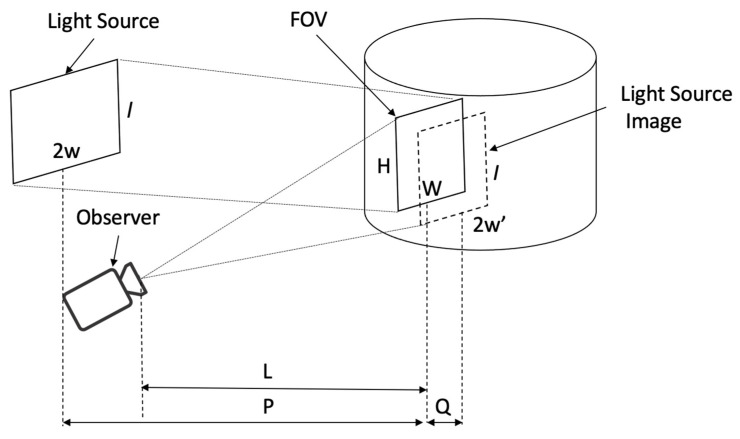
The geometry of specular reflection imaging. Distance P is critical, but Distance L does not impact the methodology (can be larger or smaller than P). Further, the observer (i.e., camera) is not aligned with the light source.

**Figure 4 sensors-22-02830-f004:**
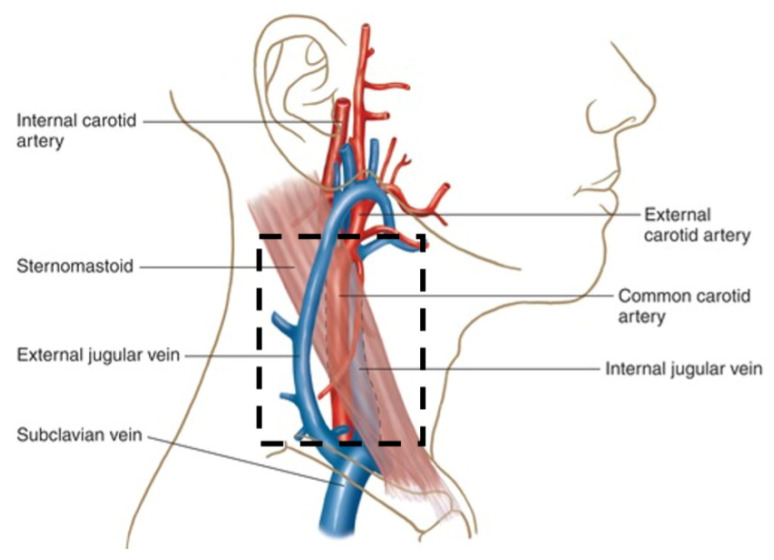
Carotid artery (CA) and Jugular Vein (JV) branches and region of interest (ROI) identified with a black rectangle.

**Figure 5 sensors-22-02830-f005:**
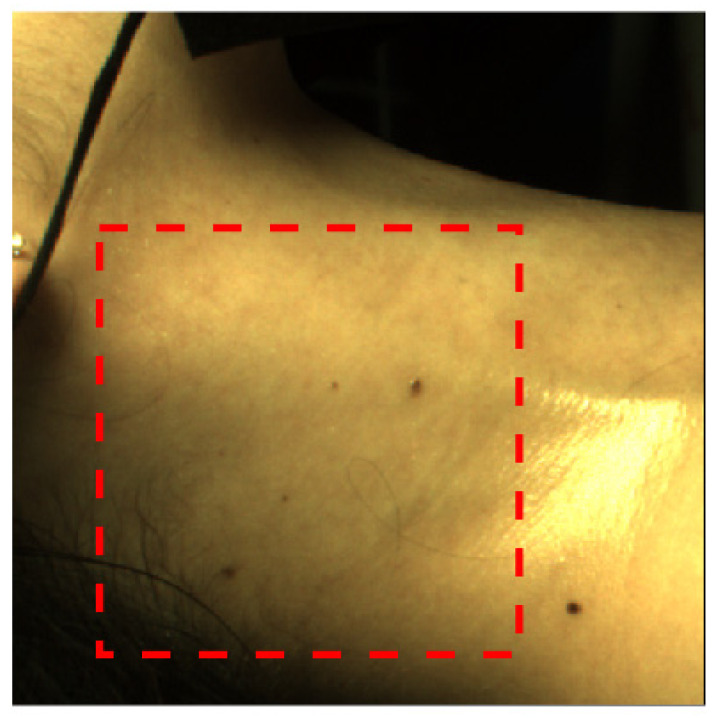
First collected frame, with the selected ROI shown in the red rectangle.

**Figure 6 sensors-22-02830-f006:**
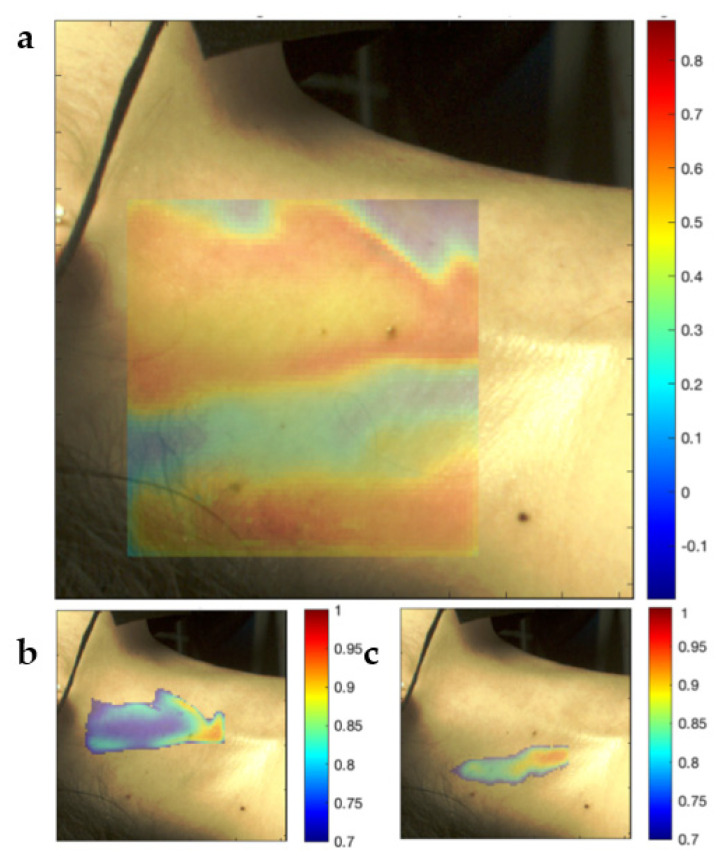
False color maps representing the Pearson correlation of local SRVI signals to the global SRVI signal (**a**) and of local SRVI signals to the average SRVI signal within high-correlation sub-regions segmented using spatial clustering (**b**,**c**).

**Figure 7 sensors-22-02830-f007:**
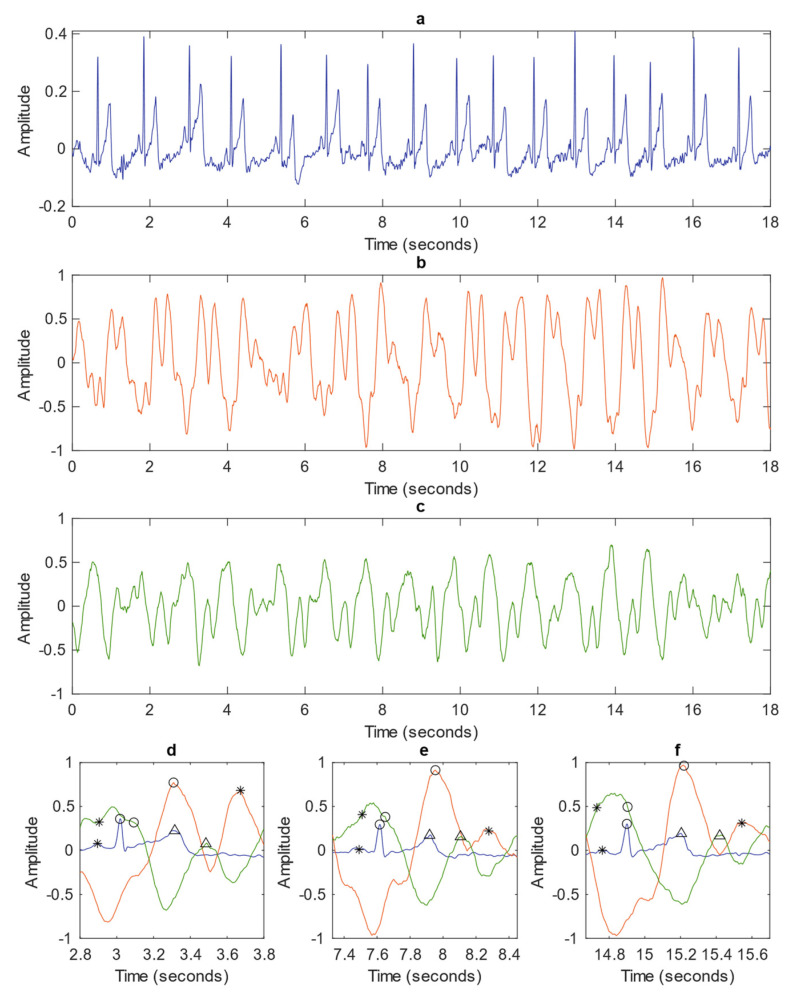
Complete duration of ECG (**a**), SRVI carotid signal (**b**), and SRVI jugular signal (**c**) captured over 18 s. Single cardiac cycles showing the synchronization of all three signal from early in the acquisition period (**d**), the middle of the period (**e**), and the end of the period (**f**). In the single cycle plots (**d**–**f**), the ECG is depicted in blue (p-wave as a star, r-peak as a circle, t-wave as a triangle), the carotid SRVI signal in red (systolic as a circle, diastolic as a star), and the jugular SRVI signal in green (a-peak as a star, c-peak as a circle, and v-peak as a triangle). Amplitude is in mV for ECG and pixel intensity for SRVI signals.

**Table 1 sensors-22-02830-t001:** Data analysis outputs.

Output	Description
Local SRVIs	SRVIs measured in 5 × 5 pixels area
Global SRVI	SRVI measured across all pixels in the ROI
Correlation False Color Map	Pearson correlation of each local SRVI to the global SVRI, shown as a false color map overlaid on the first frame
Spatial Clustering-Derived Sub-regions	Sub-regions within the ROI that exhibit internal high correlation and low correlation across sub-regions. The SRVIs are averaged within the sub-regions to generate representative signals.

## Data Availability

The video data presented in this study are available on request from the corresponding author.

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
