# Peer review of "Towards Development of Specular Reflection Vascular Imaging"

_sensors, 2022, doi:10.3390/s22082830_

Round 1

Reviewer 1 Report

The manuscript of Timothy Burton et al., with the title "Towards Development of Specular Reflection Vascular Imaging", describes the theoretical and practical validation of a new method for cardiovascular diagnosis through optical imaging.

The work presented in this manuscript, although somewhat based on other methods is innovative and the results obtained in the first test seem to validate the method.

The manuscript is well written and organized and with detailed explanation from the beginning to the end, providing a joyful reading. Authors also presented the limitations of the proposed method and future perspectives of research in this field.

I recommend the manuscript to be accepted for publication, provided that the two following corrections are made:

  1. Subsection 1.3 on line 155 should be 1.4
  2.  In line 181, the sentence should be corrected as: "flection, such measurement is problematic as a noise source..."

Reviewer 2 Report

A novel approach for non-contact detection and analysis of jugular vein's pulsations is presented and validated on a single volunteer. Paper could be published with minor revisions if the following questions/remarks are taken into consideration.

  1. Introduction could be complemented by description of typical waveforms of arterial and venous (in particular, jugular) PPG signals, i.e how they differ. It would facilitate better understanding of the further text.
  2. In Fig.3, the observer (camera) is placed at distance L from the reflecting surface and the retrangular ligh source - at distance P, presumably on the same optical axis. Consequently, some shadow of the camera on the working surface is expected that affects uniformity of illumination. Why such "orthogonal" geometry is preferred instead of using slightly tilted one (i.e. at a certain incidence angle with camera placed in the reflected light region) which could ensure better uniformity of illumination? Besides, collimated illumination beam in such case seems  favourable if compared with wide-angle LED illumination as it might reduce the role of diffuse reflection in the recorded signals. Please comment.
  3. Fig. 4 illustrates the vascular anatomy but gives no sense on distances of the blood vessels to the skin surface - this might be specified in the figure caption or in the text.
  4. Experimental setup (p.3.1). Spectral dispersion curve of the used oil could be helpful. Nothing is said on the emission spectrum of LED emitter - it is essential to assess the light penetration in skin. The geometry of experiment differs from that of Fig.3 as camera is placed behind the illuminator (150 cm and 20 cm from the target area, respectively) - or 150 cm is a misprint?
  5. Some motivation why the R, G and B color channels had to be separated from the RGB videodata would be beneficial, and why the R-channel (where maximum light penetration and, consequently, PPG signal influence could be expected) was selected for further signal analysis.
  6. Fig.6. "Heatmaps" maybe can be replaced by some other word to avoid misunderstanding that they represent thermal images.
  7. Fig.7a probably could be replaced by three separated time courses representing arterial and venous pulsations and the ECG signal; in the current representation it is hard to follow each of them. In the caption 3) obviously should be replaced by d).
  8. Page 13, line 441 - LSCI commonly means laser speckle (not scatter) contrast imaging.
